## Research Article

adolescents; mental health; rural; LMIC; photovoice

**Corresponding author:**
Ana A. Chatham;
Email: achatham@utexas.edu

# Acceptability of mental health photovoice research with adolescents in rural Mexico

Ana A. Chatham[1] 🔸, Rebecca Cook[2], Alejandro Luna[3], Patricia Vargas Espinosa[3], Karen Ramírez Calderón[4], Ivan Gutierrez[4], Yoselin Sarahi Palacios[5], Gloria Cristina Zaragoza Mendoza[5], Graciela Rivera Sanchez[5], Lizbeth Vargas Castillo[5], Brenda de la Rosa Díaz[5], Nelly Salgado de Snyder[6] and Carmen R. Valdez[7]

[1]Brown School of Social Work, Washington University, St. Louis, MO, USA; [2]Dell Medical School, The University of Texas at Austin, Austin, TX, USA; [3]Fondo Comunitario Mónica Gendreau-Fundación Comunitaria Puebla, Puebla, México; [4]Facultad de Psicologia, Benemérita Universidad Autonoma de Puebla, Puebla, México; [5]Facultad de Medicina, Benemérita Universidad Autonoma de Puebla, Puebla, México; [6]Latino Research Institute, The University of Texas at Austin College of Liberal Arts, Austin, TX, USA and [7]School of Public Health, UT Health San Antonio, San Antonio, TX, USA

## Abstract

The mental health (MH) of adolescents in low- and middle-income countries (LMIC), particularly those in rural areas, has historically been neglected in research and services, despite the documented burden MH problems represent among these populations. Settings where MH stigma is high require strategic research methods. Photovoice is a promising method for MH research in contexts of high stigma, but studies examining its acceptability with rural adolescents in LMIC remain scarce. We explored the acceptability of photovoice for MH research through perspectives of adolescents from rural Mexico who participated in a photovoice project focused on factors affecting their MH. Adolescents (n = 40) participated in focus groups where they discussed what they learned through the MH photovoice project, and the aspects of the method they perceived to be valuable. Focus groups transcripts were thematically analyzed. Participants' satisfaction with the MH photovoice project was tied to: (1) learning about the meaning, nature, and experiences of MH; (2) enjoying relationships, novelty, and fun; and (3) wishing for more time, more play, and continuity. Photovoice is an acceptable method for MH research among rural adolescents in LMIC, sparking reflection and collective dialog that can lead to the development of local initiatives.

## Impact statement

Despite the significant burden of mental health challenges, there remains limited understanding of adolescent mental health in rural global settings—compromising the development of culturally sensitive and contextually relevant interventions. Photovoice can engage adolescents in mental health research and promotion, even in contexts where stigma is high and resources are limited. In this study, adolescents from two rural communities in Mexico participated in photovoice workshops focused on mental health. The participatory and relational aspects of photovoice facilitated meaningful exploration and dialog. These findings suggest that photovoice is an acceptable and valuable method for mental health research with rural adolescents in low-resource global settings.

## Introduction

The mental health (MH) of adolescents—and the interventions designed to support them—has historically been understudied, particularly among those residing in low- and middle-income countries (LMIC) such as Mexico (Kieling et al., 2011; Adhanom Ghebreyesus, 2025; Zhang et al., 2025). This disparity is especially concerning given evidence suggesting local research drives health care provision in LMIC (Guindon et al., 2010). Furthermore, most studies on adolescent MH use surveys or questionnaires, limiting adolescents' role in research and relevance of studies (Warraitch et al., 2024; Nagata et al., 2025; Zhang et al., 2025). Participatory methods that draw on youth expertise can enhance acceptability of MH research (Warraitch et al., 2024; Nagata et al., 2025; The Lancet Child and Adolescent Health, 2025). This study explores the acceptability of photovoice, a participatory method, in terms of participant learning and the method components that enhanced value.

### Adolescent mental health in Mexico

MH problems represent a substantial burden of disease for Mexican adolescents. The Mexican Adolescent Mental Health Survey which assessed 3,005 adolescents ages 12–17 years in Mexico City indicated rates of mild, moderate, and severe mental illness were 10%, 20%, and 9%, respectively (Benjet et al., 2009). In the state of Michoacan, 9.2% of 578 adolescents surveyed reported symptoms of a probable major depressive episode, while another 20.6% reported symptoms of subthreshold depression (Jiménez Tapia et al., 2015). Suicide attempts among Mexican adolescents increased by more than 600% between 2006 and 2022 (Valdez-Santiago et al., 2023).

The well-documented role socioeconomic disadvantage plays as a social determinant of MH (Lund et al., 2018; Knifton and Inglis, 2020; Kirkbride et al., 2024) suggests that adolescents in rural Mexico may experience higher rates of mental illness than their urban counterparts, as rural areas tend to experience higher poverty than urban areas (Consejo Nacional de Evaluación de la Política de Desarrollo Social, 2023). A study in rural areas of the state of Chiapas reported that 35.8% of the 2,082 adolescent participants had symptoms consistent with depression and/or generalized anxiety disorder (GAD); among those from lower socioeconomic households, 19.5% and 29.4% screened positive for one or both disorders, respectively (Serván-Mori et al., 2021). The authors underscored how structural factors, including economic poverty and limited access to healthcare, "trigger processes leading to depression and GAD" (Serván-Mori et al., 2021, p. 186).

Like most LMIC, Mexico has insufficient funding for and providers of MH care (Pan American Health Organization, 2023; WHO, 2022). Puebla, the third poorest state in Mexico (Organization for Economic Cooperation and Development, 2022), was ranked 27th out of 32 states in access to healthcare (Organization for Economic Cooperation and Development, 2016), with rural communities experiencing the most barriers accessing MH care (Heinze et al., 2016).

Accessing the limited MH services available may carry social stigmatization, as culturally, MH care can be deemed unimportant and reserved for those who are "sick or crazy" (Wells et al., 2012). Failing to consider the role of stigma when designing MH studies may compromise findings (Parcesepe and Cabassa, 2013; Villatoro et al., 2018, 2022; Rai et al., 2023). Given these factors, photovoice, being arts-based and participatory, stood as a promising method for our research on the factors adolescents in rural Mexico experienced as affecting their MH and EW.

### Empowerment education model

The Empowerment Education Model (EEM) emphasizes the goal of education as empowerment for liberation (Freire Institute, n.d.; Wallerstein and Bernstein, 1988). Developed by Brazilian educator Paulo Freire, the EEM posits learners ought to be engaged as active co-constructors of knowledge who can examine their reality and self-determine their preferred future (Wallerstein and Bernstein, 1988). It involves listening, dialog, and action. In listening, community members learn from each other, identifying problems and priorities. Problem-posing is structured around stories, photographs, and songs and facilitated through prompts aimed to stimulate critical reflection about identified problems (Wallerstein and Bernstein, 1988). Lastly, people act towards affecting change, completing the first round of an ongoing cycle of reflection and action (Wallerstein and Bernstein, 1988).

### The photovoice method

Grounded in Freire's EEM, photovoice is a method in which participants portray their experience of a specific phenomenon through photography, engage in construction of knowledge through dialog, and take action towards positive change (Wang and Burris, 1997). While photovoice projects are meant to be adapted for best fit with audience and context, these key-elements differentiate it from other photo-based methodologies (Wang and Burris, 1997; United for Prevention in Passaic County, 2021).

Photovoice has been widely used with young people from many cultural backgrounds to assess a variety of health and social challenges including MH (Vélez-Grau, 2019; Stephens et al., 2023). Despite its increasing popularity, MH photovoice research with adolescents from rural settings in LMIC remains scarce (Stephens et al., 2023).

### Purpose

The purpose of the present study is to evaluate the acceptability of photovoice as a method for MH research and promotion through the perspectives of adolescents in rural Mexico who participated in a MH photovoice project. Acceptability was assessed in terms of the following research questions: (1) What did adolescents learn by participating in a MH photovoice project? (2) What components of the photovoice workshop contributed to their learning?

## Methods

We conducted semi-structured focus groups to learn about the experiences of adolescents who participated in a MH photovoice project. Epistemologically, our study was grounded in social constructivism which frames social interactions as shaping individuals' subjective understanding of reality (Creswell and Poth, 2017). Below we describe the photovoice project procedures, as context for the focus group findings. All activities described here were included in the same proposal approved by Institutional Review Boards at The University of Texas at Austin (UT Austin) and at the Benemérita Universidad Autónoma de Puebla (BUAP).

### Setting

The rural communities involved in this study are located in the Atlixco Valley, about an hour drive from the city of Puebla, in the state of Puebla, Mexico. Agriculture is the primary source of income for residents whose households are largely considered low-income, with approximately 87% below the poverty line (Consejo Nacional de Evaluación de la Política de Desarrollo Social, 2015). The school where this study took place serves middle and high school students from two neighboring communities with a combined population of approximately 1,300 residents. Middle school classes take place in the mornings and high school classes happen in the afternoons in the four classrooms composing the school building. Students typically walk to and from school which can take over 30 min each way. Puebla's rural communities compose the service area for Academic Model Providing Access to Care (AMPATH), a global network of universities collaborating with the public health sector to reduce health disparities (AMPATH Mexico, n.d.).

## Recruitment and consent

Using a convenience sampling strategy, all 7th–12th grade students were invited to participate in the MH photovoice project (n = 47). Participation in the workshops did not mean participation in the research. Adolescents could participate in all workshops without having their information used for research purposes, including the evaluation focus groups on which this paper is based. Research team members discussed the project in-person with adolescents and obtained signed adolescent assent prior to the workshops.

A one-page informational sheet was sent home to inform families about the study. Parents who did not wish their children to participate in the study could contact the school and / or the researchers, up to two weeks after the information was sent home, to ask questions and / or decline participation. Researchers also made themselves available at the school at set dates and times for parents to meet in-person and ask questions. This is what was deemed culturally appropriate for this setting by our community liaisons who had long-standing relationships in the community. Obtaining parental permission individually and in-person was not feasible due to the site of the study being remote and rural, and parents of potential participants spending the day out working in the fields and returning home in the evening.

Due to the stigma associated with MH in the local context, we joined scholars who use terms such as "emotional wellbeing" (EW) or "psychological wellbeing," among others, instead of or interchangeably with "mental health" when conducting studies with populations where MH stigma is high (Nastasi and Borja, 2016).

## Sample

Of the 47 adolescents who participated in the photovoice workshops, 40 participated in the focus groups. Four middle school students did not assent to participate in the research study and another three students were absent on the day focus groups were conducted. No parents declined their children's participation in the study. Among the research participants, middle school students were 13–14 years of age and more than half identified as female (68.4%). High school students were 14–17 years of age and less than half (43.4%) identified as female.

## Mental health photovoice workshops

In October 2023, we facilitated two four-session MH photovoice workshops with adolescents from two rural communities in the state of Puebla, Mexico. The primary purpose of this photovoice project was to identify the factors adolescents perceived as affecting their MH and EW. Workshop sessions lasted approximately three hours each and took place over the course of one week, during school hours at the school where the adolescents were enrolled. The first author, a trilingual (Portuguese, Spanish, English) licensed clinical social worker native of southern Brazil with previous photovoice research experience, then a doctoral student, was the primary facilitator of all activities. Four local advanced medical and two psychology social service externship students, known in Spanish as *pasantes*, from BUAP, assisted with small group activities after being trained by the main facilitator. All *pasantes* were Mexican and native-Spanish speakers; five of them were female, one male; medical students had previous experience with health promotion and services in the communities where the study took place.

Additionally, two representatives from Fondo Mónica Gendrau, a community-based organization (CBO), and AMPATH faculty and staff assisted with various tasks in the planning and implementation of the project. Workshops were organized in four phases of discovery and action: (1) framing; (2) collecting; (3) analyzing; and (4) reporting (United for Prevention in Passaic County, 2021; Solis et al., 2024b, 2024a) (see supplementary material).

*1. Framing.* Adolescents were provided with a workbook that included exercises as well as information on MH services available locally (*i.e.*, Life Line, Youth Support Line, closest health center offering mental health services). Besides the typical components of photovoice workshops (*i.e.*, photography training, safety and ethics instruction, production of narratives), we also conducted a free-listing exercise (Keddem et al., 2021) and an appreciative inquiry discussion (Cooperrider et al., 2003). Free-listing is a technique used to explore how a term is defined by people of a shared background. Participants are asked to list everything they can in relation to the topic of interest. Frequency and salience of terms listed are analyzed (Keddem et al., 2021). For this exercise, we asked participants "What comes to mind when you hear MH and EW?"

Appreciative inquiry invites participants to reflect and share about a peak experience from the past, current values, and hopes for the future (Cooperrider et al., 2003). Our questions were: (a) Tell me a story about a time when you felt that you, your family, or your community were emotionally and mentally healthy; (b) What do you value most about yourself, your family, your community? What do you value most about the story you shared? and (c) What do you wish and dream for your / your family's / your community's future in terms of being emotionally and mentally well?

*2. Collecting.* Participants were asked to take pictures in response to the following prompts: (a) What facilitates the MH and EW of youth in this community? and (b) What hinders the MH and EW of youth in this community? Groups of three to four adolescents were provided with one instant polaroid camera and film. Participants could also use their cell phones to take pictures as well as share photos they had previously taken. Each participant could take five pictures using polaroid cameras. There were no limits on the number of pictures taken with phones. Allowing flexibility around the photos was a way to share power with participants while also facilitating the collection of relevant data.

*3. Analyzing.* Following Wang and Burris' (1997) three-stage participatory analysis process—selecting, contextualizing, and codifying—participants decided which and how many pictures to contribute to the project. The final collection consisted of 179 pictures / narratives produced during three rounds of photography.

The SHOWeD method was used to guide participants' contextualization of the photos (Wang et al., 1998; United for Prevention in Passaic County, 2021). SHOWeD is an acronym that offers prompts to describe an image, based on Freire's strategy for problem posing: (1) What do you See here? (2) What is really Happening here? (3) How does this relate to Our mental health? (4) Why does this condition or situation Exist? and (5) What can we Do about it? We used an adapted version of SHOWED which excludes one of the original prompts for ease of understanding (United for Prevention in Passaic County, 2021). While participants were provided with worksheets with the SHOWeD prompts and encouraged to use them as a strategy to start drafting their narratives, they were also given freedom to write in their preferred format. Codification happened in small groups, as participants pile-sorted their pictures based on how the pictures related to each other and to the research questions. Three of the main themes generated through this activity were the natural environment (*e.g.* sunsets, plants, animals),

positive relationships (*e.g.* friends and family), and hobbies (*e.g.,* sports, music, games) (forthcoming).

**4. Reporting.** Participants curated and assembled the photos and narratives for a concluding exhibition hosted in the school. To preserve participant confidentiality, the individual creator of each piece was not specified in the exhibition. Participants received certificates after a brief presentation in which facilitators and self-selected participants provided a summary of the work to families and teachers (Figure 1). An exhibition to share the youth's work in a public site is currently under planning.

### Focus groups procedures

Focus groups took place two days after the last photovoice workshop at the school (same location as the photovoice project). Focus groups started with facilitators introducing themselves and the purpose for the group, which was to learn about adolescents' experiences with the photovoice project. Facilitators encouraged adolescents to provide honest feedback so that future projects can be improved. Facilitators asked adolescents to come up with a pseudo name, so that their answers, although recorded, would remain anonymous. To minimize social desirability bias, the main facilitators of the photovoice workshops were not present during the focus groups and adolescents were instructed to write down their answers, before sharing them verbally. A talking piece was used to promote participation and listening. Participants' written answers were collected at the end of the session.

Three of the four focus groups were led by the senior author, an established bilingual social science researcher with previous experience with youth photovoice research and focus group facilitation, and who had not been present during the photovoice workshops. Time constraints required one focus group to be facilitated by

**Table 1.** Focus group participation (n = 40)

| Focus group ID | Participants grade | Number of participants | Facilitator |
|---|---|---|---|
| FG1 | 8th | 11 | CV |
| FG2 | 9th | 8 | CV |
| FG3 | 10th | 10 | CV |
| FG4 | 11th and 12th | 13 | RC |

another well-qualified facilitator, the second author, a bilingual AMPATH administrator, established researcher, and medical faculty who played minor roles during the photovoice workshops. Questions asked in the focus groups included "What did you learn about yourself through the project?" "What did you like about the project?" and "What could have been better in the project?" Focus groups were conducted in Spanish and each lasted approximately one hour. Details about focus group participants and facilitators are included in Table 1.

### Data management

Focus groups were audio recorded with digital voice recorders. The hand-written responses to the focus group questions were typed into a Word document, separated by grade and question. Device malfunction or user error impeded the recording of one of the focus groups (FG4). Recordings of the other three focus groups were professionally transcribed in Spanish, their original language. All data and analysis files were saved in a university-approved cloud-based storage platform.

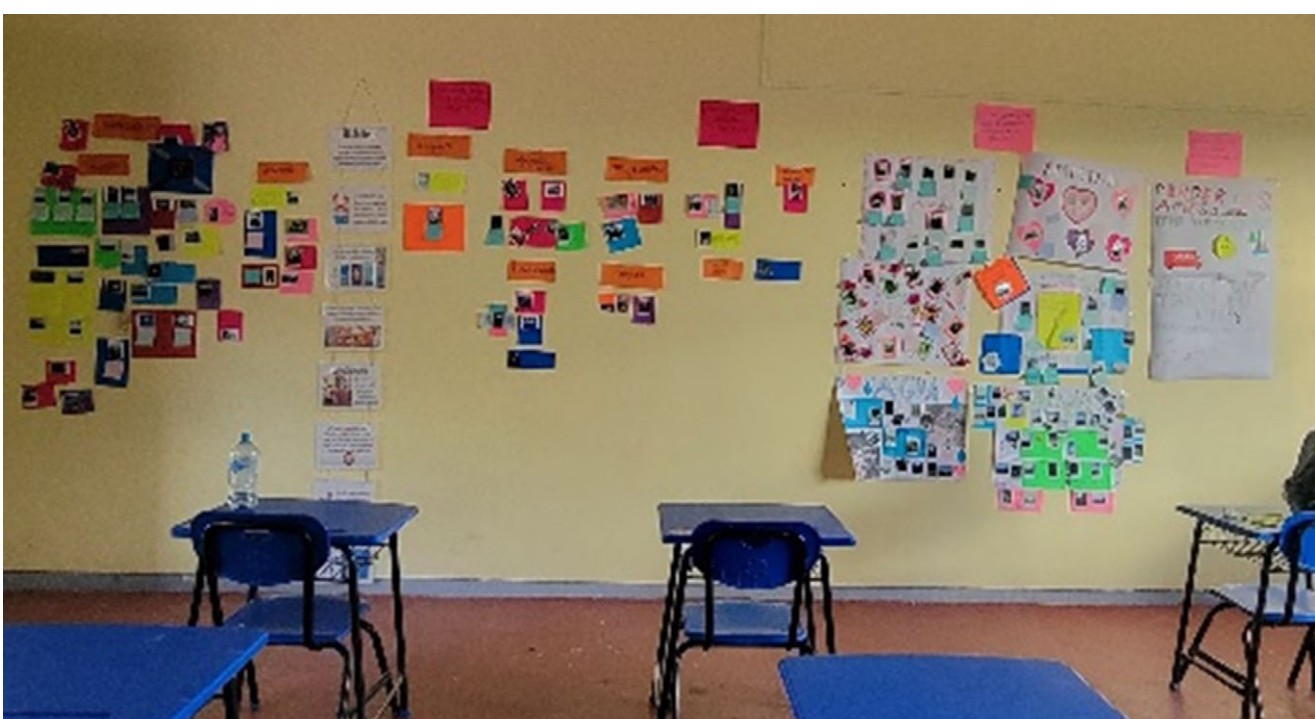

**Figure 1.** Exhibit set up by participants.

## Data analysis

The dataset utilized for this study included (1) transcripts of three of the focus groups and (2) short written answers to focus group questions. Data were analyzed in Microsoft Word using inductive thematic analysis (Guest et al., 2012) which entailed reading all transcripts and written notes, line-by-line coding, grouping individual codes into larger categories, and generating a codebook. The first author took the primary role in the coding of the transcripts. Codebook development proceeded through the iterative coding of data, revising of codes and their definitions, and discussions with members of the research team, particularly with the senior author who facilitated three of the focus groups.

To increase the transferability, dependability, and confirmability of the data analysis, data were coded in Spanish, their original language. The first author translated the quotes utilized in this manuscript and the senior author checked the translation for accuracy. Original quotes in Spanish are included as supplemental material for increased transparency and trustworthiness (Younas et al., 2021; Dolan et al., 2023).

## Reflexivity

All researchers involved in the study were either born in or have been long-term residents of Latin America and are attuned to cultural differences, even within Latin America. Researchers relied heavily on local partners and the communities to understand the rural context of Puebla, including idioms and practices. The researchers practiced reflexivity and humility to approach any differences with curiosity. The ways in which the researchers' identities and experiences impacted the design and implementation of this study, particularly relational dynamics with participants and other community members and data analysis were points of reflection and discussion.

## Findings

Analysis of focus group data led to the development of three themes: (1) learning about the meaning, nature, and experiences of MH and EW; (2) enjoying relationships, novelty, and fun; and (3) wishing for more time, more play, and continuity.

### Learning about the meaning, nature, and experiences of MH and EW

Adolescents described learning about MH and EW during the photovoice project. They specifically described learning about the meaning, nature, and personal experiences of MH and EW. Looking at their lives through the lens of MH and EW allowed adolescents a new perspective of their relationships to themselves, their families, communities, and the world. Participants learned from the workshop that part of EW can be defined as "our emotions, is what makes you happy" (middle schooler).

This high schooler described how the photovoice workshops stimulated a process of reflection towards self-awareness, starting with capturing pictures of places she liked, followed by developing awareness and language to identify and express what these places meant to her:

> I just took photos because I liked them and that's it, but later, in the workshop when they asked for the photos that we liked, that's where I learned why I took those photos of the sunsets, of the landscapes. (…) I understood that every moment, every object, every animal (…) is beautiful, it understands you, it relaxes you, it brings you peace of mind.

This type of reflection empowered participants with the realization that there are things one can do to manage one's emotional states. Although the workshops offered limited instruction around coping or emotional regulation skills, the process itself was stated to allow adolescents to identify adaptive ways to approach difficult situations. For example, one middle schooler said he gained awareness of several strategies to manage negative thoughts "being with a pet, a friend, going out to special, beautiful places or looking at photos." A high school participant reflected on how the appreciative inquiry discussion taught her to accept things she cannot change "I learned to accept my parents' separation and to know how to understand them and support them in whatever they decide to do." Another high schooler shared the workshops taught her to reframe her aunt's migration around her aunt's realizing a dream, which helped her cope better with the grief resulting from her aunt's absence.

Participants further described learning about the changeable nature of emotions as they engaged in identifying, photographing, and discussing factors that affected them. As this high school participant wrote "thanks to the photovoice workshop I learned to value things that seem simple and to relate these to an aspect in my life that is important to me, contributing to my emotional health."

Lastly, participants discussed learning about the strengths and needs of their communities and their vision for having needs met. Participants shared how the photovoice project renewed their awareness of the beauty of their surroundings and connections and their commitment to care for them, as this middle school participant wrote: "I learned many things in this photovoice project: that my community is beautiful and we must take care of our traditions of our culture, and the beautiful view we have, and nature, and the different species of animals." Participants also recognized the need for more formal community supports, highlighting the lack of MH services, youth programming, and academic guidance. As noted by this high schooler after describing various stress-inducing circumstances: "because here there is no one to talk to, there is no way to distract yourself or anything like that. Yes, more support is needed, whether in the community or here at school."

### Enjoying relationships, novelty, and fun

Adolescents characterized their experience with the photovoice project as very positive, with a few participants stating they would not change anything about it. Specifically, adolescents' positive feedback about the photovoice project was centered on (1) experience with facilitators; (2) time with friends; and (3) picture taking.

Adolescents expressed highly valuing their time with the project facilitators, explaining they enjoyed getting to know them and appreciated the way facilitators treated them. "I liked being with the people who came, getting to know them, learning with them" (middle schooler). Another high schooler reflected: "I liked it, because I had been feeling very discouraged. Other people came, I met them, I felt a sort of happiness. I think because I had not gone out in a long time." Another participant expanded on how the facilitators shared about themselves and demonstrated interest in getting to know participants. Adolescents expressed appreciating facilitators' positive regard for the participants. As this high schooler described: "they were kind, we got along well, we respected each other, they understood us, they never offended us." Other

adolescents stated they appreciated the facilitators' positive affect—"what I liked most was their joy, their smile all the time, their tranquility, their patience" (high schooler). Participants described they felt trusted by the facilitators, "they trusted us because they lent us the camera to take photos and we explained some photos and it was a nice and trusting feeling" (high schooler).

Adolescents also reported they enjoyed participating in the project and favored the small-group format because it gave them an opportunity to "enjoy moments with our friends" (middle schooler). Engaging in the process with a set group of peers of their choosing was highlighted as a positive: "I liked it when we went out to do work with peers" (high schooler).

Picture taking was also identified as a positive aspect of the project in that it prompted adolescents to notice things about themselves and their surroundings that ordinarily went unnoticed and allowed them to express this noticing to others. As one middle schooler recalled, "It had been a while since I left my house. Yes, I went out, but from the store to my house and from this house, not before—It's been a long time since I went to see." Additionally, adolescents reported enjoying the novelty of the polaroid cameras, "I liked and I was impressed to know cameras that already print photos. I just waited a few minutes and that's it" (middle schooler).

### *Wishing for more time, more play, and continuity*

Adolescents' suggestions for improving the photovoice process related to two main areas: logistics and programming. Regarding logistics, participants expressed wishing for a longer program of larger magnitude. "I would like to continue taking more photos and continue more with the photovoice project" (high schooler).

Programming wise, participants suggested that more play-based activities be incorporated in the workshops, including sports and games. More photography instruction was another recommendation, "the workshop should also include photography techniques and put them into practice" (high schooler). Moreover, participants proposed having a path forward to continue developing relationships and increasing knowledge about MH: "that they come more often, that they give new workshops like, to talk about depression" (middle schooler). Lastly, a few participants provided general comments on their desire for better communication and social climate during the workshops. One participant wrote "make us feel more trusting" (high schooler). Another, "more information (about) where they give information" (high schooler).

### Discussion

In this study, we assessed the acceptability of photovoice for MH research and promotion among adolescents in rural Mexico. Analysis of focus group transcripts and written responses led to the development of three themes describing what participants learned through the project and which components enhanced value.

The awareness and knowledge reportedly gained by participants in terms of the meaning, nature, and personal experiences of MH and EW through participation in the photovoice project support the value of photovoice as an acceptable method for MH promotion (Wang et al., 1998), in alignment with findings from other studies (Vélez-Grau, 2019; Stephens et al., 2023). Since talking about emotions, feelings, and MH was not routine for our study's participants, thinking about what affected their MH and EW required them to acknowledge (1) that MH and EW are part of their life, (2) that MH and EW are not static, and (3) that the quality of their

MH and EW is the product of factors that can be external and internal to the self. Participants' engagement with the prompts for photography naturally led them to these implicit and important realizations, suggesting photo-based strategies are conducive to strengthening emotional awareness and articulating emotional content that may be otherwise, difficult, whether due to the cultural context or the biopsychosocial dynamics characteristics of the adolescent stage. Additionally, the appreciative inquiry discussion made for a time of deep sharing and learning. A few of the reflections made by adolescents about things they learned in terms of adaptive coping happened during this activity.

Overall, participants' positive evaluation of the photovoice project, particularly the photography and relational components, are consistent with other studies (Anderson et al., 2023; Stephens et al., 2023). However, the degree and nature of the value participants attributed to the relational components, especially their reflections on how they were treated by facilitators, stand as somewhat unique. It is possible that the facilitators' approach, based on empowerment education, stood out because of its contrast with the relational dynamics typical of hierarchical cultures, in which respecting elders by listening and following directions is valued. Participants' positive response to facilitators suggests approaching relationships through a lens of empowerment is an acceptable and valued approach for engaging with adolescent participants. Another contextual factor potentially influencing how participants experienced their relationship with facilitators is how infrequently participants interact with people from outside their communities due to lack of transportation, time, and financial resources.

Participants' reflections also revealed areas for improvement in our project. Participants' recommendations regarding the inclusion of more play-based activities suggest an opportunity for better tailoring to the developmental stage of participants. Specifically, the middle school group could have benefitted from more activities that incorporated movement and play. While photography and activities such as the "MH true or false" game were interactive and playful, other activities were possibly not as well suited to the attention span and cognitive skills of middle schoolers. Expanding opportunities for creative exploration through partnerships with artists might improve the experience for younger participants, as done in a project which had a professional photographer provide guidance on storytelling through photography (Solis et al., 2024a).

Although not specifically mentioned in the focus groups, the free-listing exercise proved challenging as adolescents were hesitant to write "wrong" answers. Placing the free-listing exercise at the initial workshop was meant to capture adolescents' understanding of MH prior to any influence from the facilitators. However, it likely contributed to adolescents' hesitant engagement in the activity given the early stage of trust development and the high MH stigma in the community.

Finally, while participants' requests for additional time can be interpreted as an indication of their positive experience in the project, they also lead us to more carefully consider the implications of ending a process which was experienced as deeply relational. Scholars have highlighted that the end of a study can leave participants with a sense of loss, particularly in studies that require emotional vulnerability and relational trust (Atkinson, 2005; Northway, 2000). While our team has long-term commitments and plans for the continuation of this work, factors including funding, career trajectories, and institutional demands affect the timeliness and focus of future endeavors. We navigated this tension between honoring the (mutual) desire for continued work and the uncertainty of what that might look like by honestly disclosing this

to community members (adolescents, school staff, family members). Ideally, however, institutions and funders should better align their expectations and requirements, so that the impact of participatory research can be maximized through more timely continuation of efforts. Additionally, projects like ours might consider how to equip key community members (*e.g.* teachers, parents) to facilitate follow-up engagement and action, sustaining the project gains in the long-term.

## Limitations

This study is not without limitations. First, our sample did not include adolescents who are not in school, which is the case of many in rural Puebla. Future studies should include or focus solely on adolescents who are not in school. Second, although we were able to analyze the written answers provided by FG4 participants, the loss of the audio-recording for FG4 may render our findings incomplete. Third, power dynamics (between adolescents and facilitators and adolescents themselves) likely played a role in what and how participants shared during focus groups. We attempted to minimize power dynamics by having different facilitators for focus groups and photovoice workshops and by having participants write down answers to the focus group questions. Yet, considering what may not have been said is a useful and important exercise to inform future studies.

## Conclusion

Through this paper, we presented photovoice as an acceptable and valuable method for MH research and promotion among adolescents in rural LMIC contexts with high MH stigma. Our findings on adolescents' learning through a MH photovoice project, along with exploration of specific components that were perceived as valuable offer researchers and service providers useful guidance in how to engage adolescents.

**Open peer review.**  To view the open peer review materials for this article, please visit http://doi.org/10.1017/gmh.2025.10080.

**Supplementary material.**  The supplementary material for this article can be found at http://doi.org/10.1017/gmh.2025.10080.

**Data availability statement.**  Data can be made available upon request.

**Acknowledgements.**  We would like to thank the adolescents who participated in the study for their openness, enthusiasm, and willingness to share their experiences. We are also grateful to the school administrators and teachers as well as to the adolescents' families for their warm welcome. Lastly, we extend our thanks Dr. Laura E. Gomez Mendoza for her support.

**Author contribution.**  Ana Chatham: Conceptualization, Funding acquisition, Methodology, Data curation, Formal analysis; Writing – original draft, Project administration. Rebecca Cook: Conceptualization, Funding acquisition, Methodology, Investigation, Writing – review and editing, Supervision, Project administration. Alejandro Luna: Conceptualization, Resources, Project administration, Writing – review and editing. Patricia Vargas: Conceptualization, Resources, Project administration, Writing – review and editing. Karen Ramírez Calderón: Investigation, Writing – review and editing. Ivan Gutierrez: Investigation, Writing – review and editing. Gloria Cristina Zaragoza Mendoza: Investigation, Writing – review and editing. Graciela Rivera Sanchez: Investigation, Writing – review and editing. Lizbeth Vargas Castillo: Investigation, Writing – review and editing. Yoselin Sarahi Palacios Contreras: Investigation, Writing – review and editing. Brenda E. de la Rosa Díaz: Conceptualization, Writing – review and editing. Nelly Salgado de Snyder: Conceptualization, Writing – review and editing. Carmen R. Valdez: Conceptualization, Methodology, Investigation, Formal Analysis, Writing – review and editing, Supervision.

**Financial support.**  The project described was supported by the Lozano Long Institute of Latin American Studies' (LLILAS) ED Farmer Fellowship and AMPATH Mexico. This work was also supported by Grant Number T32MH019960 from the National Institute of Mental Health. The content is solely the responsibility of the authors and does not necessarily represent the official views of the National Institute of Mental Health or the National Institutes of Health.

**Competing interests.**  The authors have no conflicts of interest to declare.

**Ethical standard.**  This study was approved by the Institutional Review Board at The University of Texas at Austin (STUDY00004327) and at the Benemérita Universidad Autónoma de Puebla. All participants provided written informed assent.

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
