## [Reviewer Report]

Thank you for the important work on promoting mental health for adolescents in rural communities. I have a general concern about this paper. It is very confusing to read this paper, especially in the beginning. The title and abstract mistakenly present the study as a photovoice study but it was not. This issue is mainly contributed by—

1. The term “photovoice” is misused throughout the paper. Photovoice is a research design with a set blueprint and guiding conceptual frameworks. Not all photo-based projects constitute “photovoice”, nor can the blueprint of the research design be “creatively” altered. Although the photo workshops described in the paper were creative and seemed fun for the adolescent participants, no photovoice methodologies were properly reflected in the implementation of the workshops.

2. The purpose of the study was not clearly presented. Not sure if it was the original plan, but the authors seemingly meant to conduct program evaluation on the photo workshops using a research process involving focus groups. The purpose of the study was not clearly stated in abstract or Introduction; the purpose statements are also “evolving” as reading on through the paper.

It is suggested that all wording related to “photovoice” are stripped throughout this paper, the purpose of the study is clearly and consistently stated in Abstract, Introduction, and Discussion, and the title truthfully reflect what the study was about.

Abstract

The MH problem of the study was not clearly stated, nor was the study using photovoice clearly justified. It seems to me the study was to explore perceived MH problems of rural adolescents, but there was no information about what MH problems existed and how prevalent. Re-write the abstract.

Introduction

• Page 3 Lines 40-45: The statement and its source of citation are confusing. It is not clear what the message is that the authors are trying to convey.

• Page 3 Lines 45-50: This statement is premature, as the research problem under study has not yet been clearly stated, nor was it stated why it was important to conduct the study.

• Page 3 footnote: This footnote reads a critical justification for the study. The statement should be included in the texts, as opposed to be hidden as a page footnote.

• Page 4 Lines 26-31: I find this statement troubling. It implies the blanket stigma that people who are poor tend to have mental illness. Do you have data to report how factors such as lack of access to care and parental absences due to financial needs impose negative effects on adolescent MH? Also, remove the wording “well-documented link” and state how the sources of citation established the connections between poverty and mental illness.

• Page 4 Lines 33-35: Another troubling statement- “the poorest state with highest percentage of indigenous populations”. Remove this portion unless the cited study confirmed the relationships that poverty and indigenous backgrounds themselves were the sole/direct causes of high prevalence of mental illness and suicide attempts among adolescents.

• Page 5 heading “The Photovoice Method”: It is unusual to describe study method in Introduction section. At the most, use one sentence to briefly describe the method in Introduction. Move the rest of the details to Methods section.

• Page 6 heading “Mental Health Photovoice Workshops”: This section has too much details to be included under Introduction. It is very rare to provide this level of details in Introduction section. As this paper was to conduct a program evaluation and participants were recruited before the workshops, the workshops were part of the study- the description of the workshops should be listed under Methods to allow the due scrutiny on its implementation. Move the contents in this section to the Study Procedures section under Methods. Furthermore, the description here is misleading to indicate this was a photovoice study but in fact, it was not. The implementation described in this section showed major deviations from the authentic photovoice methodology, e.g., there was no conceptual framework to guide the use of this method; the original SHOWED questions were not used in the project, and there is no description how the research team used the questions to facilitate group discussion besides the questions were given to students to write their narratives. At the most, this can only be described as a photo-based project. The term photovoice is misused in this paper. All the wordings related to “photovoice” in this paper must be stripped.

• Page 6 Lines 17-25: Describe the characteristics of these schools and students, to allow examining if they properly represented rural LMICs. Also, describe whether these workshops took place during school hours or after school.

• Page 6 Lines 38-43: Describe how individual student’s voice was preserved during the group processes and none was further marginalized. Add a statement whether the stated workshops were also approved by the same IRB entity that approved the focus group study.

• Page 7 Lines 21-24: Describe the reason you allowed students to take photos in these different ways and what you did to address any social comparisons and peer pressures during the process. Describe the consent process for this level of sharing (esp. previously taken photos) and whether parental consent was obtained related to such level of sharing.

• Page 7 Line 33 “The SHOWeD method”: The description in this section has multiple serious deviations from how the method is supposed to be done. These questions are not the original SHOWED questions.

• Page 7 Lines 42-49: Did any student provide narratives without answering any of the proposed questions? What was the percentage these questions were answered in student narratives? Describe how the research team facilitated group discussions by using the proposed questions. It is not clear if the SHOWED method was properly implemented in the project. Lastly, describe how the photo piles were sorted and if all photos were included into one of the piles. Report the average percentage of photos taken by individual students were pile-sorted.

• Page 7 Lines 51-54: What was done to protect confidentiality? State whether approval was obtained from an IRB entity for the concluding exhibition. What advocacy work did you do during Reporting to promote for example, parent awareness of MH problems, especially in this cultural contexts that you were not able to obtain parental consent?

• Page 8 Lines 10-15: This paper is very confusing because this is where it finally tells you it was not a photovoice study. The statement here sounds like the authors meant to conduct a program evaluation to examine the impact of the school MH project. Provide information to answer the following questions. One, was “this present study” approved by the IRB before the photo workshops were implemented? Two, was “this present study” already planned to be conducted before the photo workshops being implemented? The study results would be questionable if the answer to any of the above questions is No, because it seemed the authors were trying to “harvest” input about a project that was already deemed “great”, and student participants of this present study could be biased by all the ceremonies and any publicities for the project.

• Page 8 Line 15 “The following research questions”: The questions were more appropriate for the discussion guide of your focus groups. What exactly was your research question given the stated purpose of the study?

Methods

• Page 8 Paragraph 1 under “Methods”: Add the heading “Study Design” for this paragraph. Also, add description about the qualitative research framework that guided your study.

• Page 8 Lines 48-50: What were the average family income for these two neighboring communities?

• Page 9 Lines 12-15: A great portion of the findings reported in this paper, esp. Themes #1 and #2, indicates the focus groups were conducted for impact evaluation, not process evaluation. Describe what you meant by “process evaluation” and what was communicated with the participants about the purpose of the study.

• Page 9 heading “Sample”: “Participants”. State the inclusion and exclusion criteria, and how the sample size was determined.

• Page 10 heading “Focus Groups Procedures”: “Study Procedure”. Include the description of the workshops here.

• Page 10 Lines 5-8: As the focus group was a special activity as opposed to a regular at school, describe the communications made to ensure students fully understand the intent of the focus group, so that they were not biased to express what was expected about a “special” project.

• Page 10 Lines 22-24: Describe the reason one of the groups was led by a different facilitator, any impact on the group process and results of group discussion, and what you did to address the impact.

• Page 10 Lines 26-31: Describe which aspect of program evaluation (progress vs impact) these questions were focusing on, and what the research team did to stay focused when group discussions drifted away from the questions.

• Page 11 heading “Data Analysis”: Describe if any computer software was used to aid the data analysis.

• Page 11 Line 10 “social constructivism”: Is this the qualitative research framework that guided your study? If yes, include it with a brief description in the Study Design section under Methods.

• Page 11 Lines 17-19 “discussions with members of the research team”: Provide a clearer picture how data analysis was done among members of the research team. How did the team discussion take place (e.g., online or in-person; how many meetings; whether all attended all the iteration processes), and how did the team resolve any discrepancies and achieve consensus?

• Page 11 Lines 19-22: Be more specific about which aspect of trustworthiness was enhanced by preserving the language during coding.

Results

• Page 11 heading “Findings”: “Results”

• Page 11 Lind 28 “Analysis of focus group data”: How many codes emerged, and with how many references highlighted/ identified in the transcripts and typed notes? Do you have any percentages of agreement for coding to report?

• Page 12 Lines 12-20: Suggest that all quotes in the original language are removed to a supplement document if appropriate.

• Page 15 heading “Wishing for More Time, More Play, and Continuity”: Did the workshops take place during school hours or after school? It could sound like the students wanted to get out of school work and just have fun with a group of non-school facilitators, which may not be appropriate for the school’s mission nor show promising sustainability of the project.

Discussion

• Page 16 Paragraph 1 under “Discussion”: The 1st sentence will need to be re-worked once revisions related to the purpose and methods are done.

• Page 16 Line 44 “acceptability”: This purpose about the study was never mentioned in the previous sections. Be clear and consistent about the purpose statement.

• Page 16 Line 44-47 “photovoice for MH research and promotion”: I think you meant to say the workshops. Judging from the questions asked to the participants, it did not appear the study evaluated the students' acceptability about “research”. This wording should be used more carefully throughout this paper to allow precise communication during research dissemination.

• Page 17 Lines 12-22: This long sentence is difficult to understand. The three requirements for acknowledging were not clearly reported in the Results section, which could negatively impact on the trustworthiness of this paper. Include (and elaborate) this statement in the Results section, or remove this statement.

• Page 17 Line 22 “naturally”: Consider discuss from the developmental standpoint about the likely challenges adolescents face in articulating their emotions and feelings at this age, and how photography provides a platform to facilitate the development of related skills.

• Page 17 Lines 42-45 “the facilitators’ approach, based on empowerment education”: Was the facilitators’ approach following any conceptual framework that was guiding this project? Describe the facilitators training backgrounds, including education and project training.

• Page 17 Line 52 “there relationship with facilitators”: Provide your evaluation on the sustainability of the project given the relationship and its impact on students as well as school staff.

• Page 18 Line 12 “Participants’ reflections also revealed the limitations”: Move this paragraph to Limitation, and add what can be done differently in the future.

• Page 18 Line 26 “… not as well suited to…”: This seems to suggest photovoice is not age-appropriate for adolescents, which is not consistent with the literature. Please elaborate.

• Page 18 Lines 26-31: Consider comparing the characteristics of the study participants with those in photo- or art-based studies for adolescents.

• Page 18 Lines 38-40: Move this paragraph to Limitations, and add what can be done differently in the future.

• Page 19 Lines 15-20 “factors including…”: Consider discuss program sustainability and empowering school staff, students, and even parents in the future.

Conclusion

• Position the section by focusing on what the workshops had achieved and what need to be done differently in the future.

References

• Provide a translated title in English for each of the non-English citations.

---

## [Reviewer Report]

Thank you for the opportunity to read this interesting study. I found the research to be relevant, clear, and useful for the academic and scientific community. One area for further strengthening the study (method section) would be to provide more detailed information on the research team’s approach to rigor and reflexivity. Specifically, elaborating on how the team acknowledged and managed their own biases, assumptions, and subjectivity throughout the research process would enhance transparency and demonstrate a commitment to methodological rigor.

---

## [Reviewer Report]

Thanks for the opportunity to review this paper. This is an interesting paper where the authors have examined the feasibility of using Photovoice methodology in conducting mental health research among adolescents in low resource settings. As participatory approaches as Photovoice are gaining traction in mental health research, this paper can be an important contribution in guiding future researchers on how to do this. However, I feel the paper could be significantly improved. Here are my specific comments –

Introduction section

1. In the Intro section, can the authors look for more recent citations? E.g. Kieling’s paper from 2011 can be updated with more recent systematic reviews. Link between poverty and mental illness also has more recent citations.

2. I am confused about how the authors have stated the goal of the papers in different sections – e.g. Page 3 Line 45, they mention the study helps address the needs for …… by capturing their meanings and lived experience through participatory method. Reading through the paper, I feel that is not what the paper is intending to do. This paper is more focused on accessing the acceptability/feasibility of the PhotoVoice process, NOT the outcome i.e. their experience of mental health, which is very important in its own terms.

3. Minor comments on citations – you can ignore details like (most recently available data) pg 4 line 10.

4. Pg 4 line 40. The last line might be unnecessary as that is not the focus of the paper (though important) and causes unnecessary distraction to the readers.

5. Pg 5 – line 15 – please add citation to the statement. Possible citations include works of Yang, Kohrt. Also, authors can cite works that has used Photovoice in mental health research and intervention.

6. Overall, the introduction section can be tighter focusing more on what the paper intends to do – i.e. describe and evaluate the process of Photovoice among adolescents to do mental health research and intervention. In its current shape, it is describing more on background on MH among adolescents in Mexico (except the first paragraph in page 5). Please consider restructuring this section.

Method section

1. The authors are requested to provide more details on the PhotoVoice method OR if there are any existing papers that describe this work, please refer/cite that. For example – in the PV workshop it is not clear if there were no prompts provided to the adolescents for taking picture, free listing and pile sorting. Only later in the discussion, they mention it was something related to their understanding of mental health. Please add that. E.g. adolescents were asked to free list items/domains on XXX. The adolescents conducted pile sorting of their photos based on XXXXX.

2. Collecting – Was there any set rules for number of photos, selecting the photos they were taken?

3. Can the authors talk more about the exhibition piece, esp. in the light of confidentiality? In Photovoice projects, the participants often hesitate to share their photos publicly or add a layer to protect their privacy. Did the researchers experience anything such esp. since I am assuming this is a rural close-knit community and the exhibition took place in the school. This would be an interesting piece to add as the paper is focused on feasibility of the process.

4. Please add more details into data analysis – esp. the codebook development. Was it just the first author or others were also involved in the process. If yes, can you describe the IRR process? Did they use any coding software - Nvivo, Dedoose? Since the first author coded and it was coded directly in Spanish can you talk about the first author’s fluency? This was confusing as the author also mentions that co-author CV checked for translation accuracy.

5. Were the coders sensitive towards local culture and idioms? (since this was done directly in Spanish). How did the analysis team handle this?

Findings

1. Details like adolescents learning about MH and EW can be moved to the methods section to describe the content of the process.

2. Can the authors also talk if there were any referral process set-up for adolescents in case there were concerns regarding their safety or if they required mental health care? (esp. since the authors mention that there was limited instruction around coping and emotional regulation skills)

Discussion

1. Can the authors talk more about the facilitator’s background since you describe on relationship between participants and the facilitators? What kind of prior PV training/experience did they had? What are your suggestions for training components - things like cultural sensitivity, gender, etc that you would recommend if someone wanted to replicate this?

2. Minor comments: Pg 19, line 26 – …… funders would or should?

Overall, great paper and enjoyed reviewing it.

---

## [Editor Report]

Dear Authors,

We have received comments from three reviewers, and all agree that this is a valuable work, but it requires substantial improvements. An important aspect is that you better describe and justify the aspects of the method described. Therefore, we ask you to review the comments thoroughly and improve your manuscript, before resubmitting it.

Reviewer 1:

I found the research to be relevant, clear, and useful for the academic and scientific community. One area for further strengthening the study (method section) would be to provide more detailed information on the research team’s approach to rigor and reflexivity. Specifically, elaborating on how the team acknowledged and managed their own biases, assumptions, and subjectivity throughout the research process would enhance transparency and demonstrate a commitment to methodological rigor. 

Reviewer 2:

This is an interesting paper where the authors have examined the feasibility of using Photovoice methodology in conducting mental health research among adolescents in low resource settings. As participatory approaches as Photovoice are gaining traction in mental health research, this paper can be an important contribution in guiding future researchers on how to do this. However, I feel the paper could be significantly improved. Here are my specific comments –

Introduction section

1. In the Intro section, can the authors look for more recent citations? E.g. Kieling’s paper from 2011 can be updated with more recent systematic reviews. Link between poverty and mental illness also has more recent citations.

2. I am confused about how the authors have stated the goal of the papers in different sections – e.g. Page 3 Line 45, they mention the study helps address the needs for …… by capturing their meanings and lived experience through participatory method. Reading through the paper, I feel that is not what the paper is intending to do. This paper is more focused on accessing the acceptability/feasibility of the PhotoVoice process, NOT the outcome i.e. their experience of mental health, which is very important in its own terms.

3. Minor comments on citations – you can ignore details like (most recently available data) pg 4 line 10.

4. Pg 4 line 40. The last line might be unnecessary as that is not the focus of the paper (though important) and causes unnecessary distraction to the readers.

5. Pg 5 – line 15 – please add citation to the statement. Possible citations include works of Yang, Kohrt. Also, authors can cite works that has used Photovoice in mental health research and intervention.

6. Overall, the introduction section can be tighter focusing more on what the paper intends to do – i.e. describe and evaluate the process of Photovoice among adolescents to do mental health research and intervention. In its current shape, it is describing more on background on MH among adolescents in Mexico (except the first paragraph in page 5). Please consider restructuring this section.

Method section

1. The authors are requested to provide more details on the PhotoVoice method OR if there are any existing papers that describe this work, please refer/cite that. For example – in the PV workshop it is not clear if there were no prompts provided to the adolescents for taking picture, free listing and pile sorting. Only later in the discussion, they mention it was something related to their understanding of mental health. Please add that. E.g. adolescents were asked to free list items/domains on XXX. The adolescents conducted pile sorting of their photos based on XXXXX.

2. Collecting – Was there any set rules for number of photos, selecting the photos they were taken?

3. Can the authors talk more about the exhibition piece, esp. in the light of confidentiality? In Photovoice projects, the participants often hesitate to share their photos publicly or add a layer to protect their privacy. Did the researchers experience anything such esp. since I am assuming this is a rural close-knit community and the exhibition took place in the school. This would be an interesting piece to add as the paper is focused on feasibility of the process.

4. Please add more details into data analysis – esp. the codebook development. Was it just the first author or others were also involved in the process. If yes, can you describe the IRR process? Did they use any coding software - Nvivo, Dedoose? Since the first author coded and it was coded directly in Spanish can you talk about the first author’s fluency? This was confusing as the author also mentions that co-author CV checked for translation accuracy.

5. Were the coders sensitive towards local culture and idioms? (since this was done directly in Spanish). How did the analysis team handle this?

Findings

1. Details like adolescents learning about MH and EW can be moved to the methods section to describe the content of the process.

2. Can the authors also talk if there were any referral process set-up for adolescents in case there were concerns regarding their safety or if they required mental health care? (esp. since the authors mention that there was limited instruction around coping and emotional regulation skills)

Discussion

1. Can the authors talk more about the facilitator’s background since you describe on relationship between participants and the facilitators? What kind of prior PV training/experience did they had? What are your suggestions for training components - things like cultural sensitivity, gender, etc that you would recommend if someone wanted to replicate this?

2. Minor comments: Pg 19, line 26 – …… funders would or should?

Reviewer 3:

I have a general concern about this paper. It is very confusing to read this paper, especially in the beginning. The title and abstract mistakenly present the study as a photovoice study but it was not. This issue is mainly contributed by—

1. The term “photovoice” is misused throughout the paper. Photovoice is a research design with a set blueprint and guiding conceptual frameworks. Not all photo-based projects constitute “photovoice”, nor can the blueprint of the research design be “creatively” altered. Although the photo workshops described in the paper were creative and seemed fun for the adolescent participants, no photovoice methodologies were properly reflected in the implementation of the workshops.

2. The purpose of the study was not clearly presented. Not sure if it was the original plan, but the authors seemingly meant to conduct program evaluation on the photo workshops using a research process involving focus groups. The purpose of the study was not clearly stated in abstract or Introduction; the purpose statements are also “evolving” as reading on through the paper.

It is suggested that all wording related to “photovoice” are stripped throughout this paper, the purpose of the study is clearly and consistently stated in Abstract, Introduction, and Discussion, and the title truthfully reflect what the study was about.

Abstract

The MH problem of the study was not clearly stated, nor was the study using photovoice clearly justified. It seems to me the study was to explore perceived MH problems of rural adolescents, but there was no information about what MH problems existed and how prevalent. Re-write the abstract.

Introduction

• Page 3 Lines 40-45: The statement and its source of citation are confusing. It is not clear what the message is that the authors are trying to convey.

• Page 3 Lines 45-50: This statement is premature, as the research problem under study has not yet been clearly stated, nor was it stated why it was important to conduct the study.

• Page 3 footnote: This footnote reads a critical justification for the study. The statement should be included in the texts, as opposed to be hidden as a page footnote.

• Page 4 Lines 26-31: I find this statement troubling. It implies the blanket stigma that people who are poor tend to have mental illness. Do you have data to report how factors such as lack of access to care and parental absences due to financial needs impose negative effects on adolescent MH? Also, remove the wording “well-documented link” and state how the sources of citation established the connections between poverty and mental illness.

• Page 4 Lines 33-35: Another troubling statement- “the poorest state with highest percentage of indigenous populations”. Remove this portion unless the cited study confirmed the relationships that poverty and indigenous backgrounds themselves were the sole/direct causes of high prevalence of mental illness and suicide attempts among adolescents.

• Page 5 heading “The Photovoice Method”: It is unusual to describe study method in Introduction section. At the most, use one sentence to briefly describe the method in Introduction. Move the rest of the details to Methods section.

• Page 6 heading “Mental Health Photovoice Workshops”: This section has too much details to be included under Introduction. It is very rare to provide this level of details in Introduction section. As this paper was to conduct a program evaluation and participants were recruited before the workshops, the workshops were part of the study- the description of the workshops should be listed under Methods to allow the due scrutiny on its implementation. Move the contents in this section to the Study Procedures section under Methods. Furthermore, the description here is misleading to indicate this was a photovoice study but in fact, it was not. The implementation described in this section showed major deviations from the authentic photovoice methodology, e.g., there was no conceptual framework to guide the use of this method; the original SHOWED questions were not used in the project, and there is no description how the research team used the questions to facilitate group discussion besides the questions were given to students to write their narratives. At the most, this can only be described as a photo-based project. The term photovoice is misused in this paper. All the wordings related to “photovoice” in this paper must be stripped.

• Page 6 Lines 17-25: Describe the characteristics of these schools and students, to allow examining if they properly represented rural LMICs. Also, describe whether these workshops took place during school hours or after school.

• Page 6 Lines 38-43: Describe how individual student’s voice was preserved during the group processes and none was further marginalized. Add a statement whether the stated workshops were also approved by the same IRB entity that approved the focus group study.

• Page 7 Lines 21-24: Describe the reason you allowed students to take photos in these different ways and what you did to address any social comparisons and peer pressures during the process. Describe the consent process for this level of sharing (esp. previously taken photos) and whether parental consent was obtained related to such level of sharing.

• Page 7 Line 33 “The SHOWeD method”: The description in this section has multiple serious deviations from how the method is supposed to be done. These questions are not the original SHOWED questions.

• Page 7 Lines 42-49: Did any student provide narratives without answering any of the proposed questions? What was the percentage these questions were answered in student narratives? Describe how the research team facilitated group discussions by using the proposed questions. It is not clear if the SHOWED method was properly implemented in the project. Lastly, describe how the photo piles were sorted and if all photos were included into one of the piles. Report the average percentage of photos taken by individual students were pile-sorted.

• Page 7 Lines 51-54: What was done to protect confidentiality? State whether approval was obtained from an IRB entity for the concluding exhibition. What advocacy work did you do during Reporting to promote for example, parent awareness of MH problems, especially in this cultural contexts that you were not able to obtain parental consent?

• Page 8 Lines 10-15: This paper is very confusing because this is where it finally tells you it was not a photovoice study. The statement here sounds like the authors meant to conduct a program evaluation to examine the impact of the school MH project. Provide information to answer the following questions. One, was “this present study” approved by the IRB before the photo workshops were implemented? Two, was “this present study” already planned to be conducted before the photo workshops being implemented? The study results would be questionable if the answer to any of the above questions is No, because it seemed the authors were trying to “harvest” input about a project that was already deemed “great”, and student participants of this present study could be biased by all the ceremonies and any publicities for the project.

• Page 8 Line 15 “The following research questions”: The questions were more appropriate for the discussion guide of your focus groups. What exactly was your research question given the stated purpose of the study?

Methods

• Page 8 Paragraph 1 under “Methods”: Add the heading “Study Design” for this paragraph. Also, add description about the qualitative research framework that guided your study.

• Page 8 Lines 48-50: What were the average family income for these two neighboring communities?

• Page 9 Lines 12-15: A great portion of the findings reported in this paper, esp. Themes #1 and #2, indicates the focus groups were conducted for impact evaluation, not process evaluation. Describe what you meant by “process evaluation” and what was communicated with the participants about the purpose of the study.

• Page 9 heading “Sample”: “Participants”. State the inclusion and exclusion criteria, and how the sample size was determined.

• Page 10 heading “Focus Groups Procedures”: “Study Procedure”. Include the description of the workshops here.

• Page 10 Lines 5-8: As the focus group was a special activity as opposed to a regular at school, describe the communications made to ensure students fully understand the intent of the focus group, so that they were not biased to express what was expected about a “special” project.

• Page 10 Lines 22-24: Describe the reason one of the groups was led by a different facilitator, any impact on the group process and results of group discussion, and what you did to address the impact.

• Page 10 Lines 26-31: Describe which aspect of program evaluation (progress vs impact) these questions were focusing on, and what the research team did to stay focused when group discussions drifted away from the questions.

• Page 11 heading “Data Analysis”: Describe if any computer software was used to aid the data analysis.

• Page 11 Line 10 “social constructivism”: Is this the qualitative research framework that guided your study? If yes, include it with a brief description in the Study Design section under Methods.

• Page 11 Lines 17-19 “discussions with members of the research team”: Provide a clearer picture how data analysis was done among members of the research team. How did the team discussion take place (e.g., online or in-person; how many meetings; whether all attended all the iteration processes), and how did the team resolve any discrepancies and achieve consensus?

• Page 11 Lines 19-22: Be more specific about which aspect of trustworthiness was enhanced by preserving the language during coding.

Results

• Page 11 heading “Findings”: “Results”

• Page 11 Lind 28 “Analysis of focus group data”: How many codes emerged, and with how many references highlighted/ identified in the transcripts and typed notes? Do you have any percentages of agreement for coding to report?

• Page 12 Lines 12-20: Suggest that all quotes in the original language are removed to a supplement document if appropriate.

• Page 15 heading “Wishing for More Time, More Play, and Continuity”: Did the workshops take place during school hours or after school? It could sound like the students wanted to get out of school work and just have fun with a group of non-school facilitators, which may not be appropriate for the school’s mission nor show promising sustainability of the project.

Discussion

• Page 16 Paragraph 1 under “Discussion”: The 1st sentence will need to be re-worked once revisions related to the purpose and methods are done.

• Page 16 Line 44 “acceptability”: This purpose about the study was never mentioned in the previous sections. Be clear and consistent about the purpose statement.

• Page 16 Line 44-47 “photovoice for MH research and promotion”: I think you meant to say the workshops. Judging from the questions asked to the participants, it did not appear the study evaluated the students' acceptability about “research”. This wording should be used more carefully throughout this paper to allow precise communication during research dissemination.

• Page 17 Lines 12-22: This long sentence is difficult to understand. The three requirements for acknowledging were not clearly reported in the Results section, which could negatively impact on the trustworthiness of this paper. Include (and elaborate) this statement in the Results section, or remove this statement.

• Page 17 Line 22 “naturally”: Consider discuss from the developmental standpoint about the likely challenges adolescents face in articulating their emotions and feelings at this age, and how photography provides a platform to facilitate the development of related skills.

• Page 17 Lines 42-45 “the facilitators’ approach, based on empowerment education”: Was the facilitators’ approach following any conceptual framework that was guiding this project? Describe the facilitators training backgrounds, including education and project training.

• Page 17 Line 52 “there relationship with facilitators”: Provide your evaluation on the sustainability of the project given the relationship and its impact on students as well as school staff.

• Page 18 Line 12 “Participants’ reflections also revealed the limitations”: Move this paragraph to Limitation, and add what can be done differently in the future.

• Page 18 Line 26 “… not as well suited to…”: This seems to suggest photovoice is not age-appropriate for adolescents, which is not consistent with the literature. Please elaborate.

• Page 18 Lines 26-31: Consider comparing the characteristics of the study participants with those in photo- or art-based studies for adolescents.

• Page 18 Lines 38-40: Move this paragraph to Limitations, and add what can be done differently in the future.

• Page 19 Lines 15-20 “factors including…”: Consider discuss program sustainability and empowering school staff, students, and even parents in the future.

Conclusion

• Position the section by focusing on what the workshops had achieved and what need to be done differently in the future.

References

• Provide a translated title in English for each of the non-English citations.

---

## [Reviewer Report]

The revised manuscript demonstrates improvements. The Photovoice methodology is now more clearly articulated with the voice of adolescents.

---

## [Editor Report]

Dear authors, after review by the peer reviewers, it has been decided to accept the manuscript under its current terms.

We appreciate your time.